# Perfusable System Using Porous Collagen Gel Scaffold Actively Provides Fresh Culture Media to a Cultured 3D Tissue

**DOI:** 10.3390/ijms22136780

**Published:** 2021-06-24

**Authors:** Chikahiro Imashiro, Kai Yamasaki, Ryu-ichiro Tanaka, Yusuke Tobe, Katsuhisa Sakaguchi, Tatsuya Shimizu

**Affiliations:** 1Institute of Advanced Biomedical Engineering and Science, Tokyo Women’s Medical University, TWIns, Tokyo 162-8666, Japan; imashiro.chikahiro@twmu.ac.jp (C.I.); tanaka.ryuichiro_1@twmu.ac.jp (R.-i.T.); shimizu.tatsuya@twmu.ac.jp (T.S.); 2Department of Integrative Bioscience and Biomedical Engineering, Graduate School of Advanced Science and Engineering, Waseda University, TWIns, Tokyo 162-8480, Japan; kaiyamasaki.ky@gmail.com; 3School of Creative Science and Engineering, Faculty of Science and Engineering, Waseda University, TWIns, Tokyo 162-8480, Japan; yusuke7tobe@aoni.waseda.jp

**Keywords:** 3D tissue culture, cell sheet technology, perfusion system, convection-diffusion equation

## Abstract

Culturing three-dimensional (3D) tissues with an appropriate microenvironment is a critical and fundamental technology in broad areas of cutting-edge bioengineering research. In addition, many technologies have engineered tissue functions. However, an effective system for transporting nutrients, waste, or oxygen to affect the functions of cell tissues has not been reported. In this study, we introduce a novel system that employs diffusion and convection to enhance transportation. To demonstrate the concept of the proposed system, three layers of normal human dermal fibroblast cell sheets are used as a model tissue, which is cultured on a general dish or porous collagen scaffold with perfusable channels for three days with and without the perfusion of culture media in the scaffold. The results show that the viability of the cell tissue was improved by the developed system. Furthermore, glucose consumption, lactate production, and oxygen transport to the tissues were increased, which might improve the viability of tissues. However, mechanical stress in the proposed system did not cause damage or unintentional functional changes in the cultured tissue. We believe that the introduced culturing system potentially suggests a novel standard for 3D cell cultures.

## 1. Introduction

In recent years, there has been a rapid increase in the number of bioengineering studies using cultured cell tissues, such as regenerative medicine, drug screening systems in vitro, and the fabrication of cell-based food [1,2,3,4]. As a common and important technology in such fields, culture technologies required to realize physiological conditions and to develop valuable cell tissues have been investigated, and the fabrication of three-dimensional (3D) cell tissues has attracted considerable attention [5,6,7]. Although many trials and strategies including 3D bioprinters and cell-laden microfibers have been developed to fabricate 3D tissue [8,9], cell sheet technology has emerged as one of the most promising of these methods [10]. The main advantage of this technology is the fabrication of dense cell tissue without any inclusion inside 3D cell tissues. Thus, cell sheet technology permits the fabrication of dense tissues such as the cornea, skeletal muscle, myocardium, and other organs, such as the liver [11,12,13,14]. Many trials have been conducted in fields such as regenerative medicine, drug screening based on cell sheet technologies [15,16,17,18,19].

To improve the function of 3D cell tissue based on cell sheet technology, many strategies have been reported to simulate physiological conditions. To develop engineered cell tissues, several types of mechanical stresses have been applied to them [16,20,21]. The modification of culture media is one of the most conventional approaches in cell culture technology, requiring knowledge of chemistry and molecular biology. Furthermore, it has been reported that in vitro static culture creates oxygen and nutrient gradients in the culture media and even in the tissue [22]. In the worst case, cell tissues undergo necrosis owing to a lack of oxygen, nutrients, or accumulated waste products [23]. In addition, the constitution of culture media around the cell tissue varies during culture [24] and may result in a decrease in cell function. Thus, there is a need for appropriate technology for the replacement of the culture media around the cell tissue in vitro in order to maintain culture conditions.

In general, the static culture on a plastic culture dish is a conventional culture method. Cells comprising a cell tissue intake nutrients and oxygen from and output waste to culture media in vitro [22], which creates a gradient in the concentration of chemical substances and oxygen in the culture media. However, in addition to static conditions generating the gradient, the plastic culture surface to which the tissue adheres does not facilitate such an exchange (see Figure 1A,C) [25]. This is a significant difference in comparison to the in vivo case, where blood or lymph always exchange substances with cells. To realize such an exchange even at the bottom of the tissue (see Figure 1B), cell tissues were cultured on a cell culture insert having multiple micropores, resulting in the enhancement of the exchange and cell viability [25]. Thus, as a next step, we should realize the active replacement of the media to prevent the development of the gradient in culture media and tissues. There are two possible approaches for the active replacement of media: enhancing diffusion and utilizing convection–diffusion. The diffusion equation is expressed as follows:
(1)$$\frac{\partial \phi }{\partial t}=\nabla \cdot \left(D\left(\phi ,\;\vec{r},\;t\right)\nabla \phi \left(\vec{r},\;t\right)\right)$$
where ϕ represents the density of the diffusing material at location $\vec{r}$ and time *t*, and *D* is the diffusion coefficient. Therefore, the diffusion coefficient should increase to enhance the diffusion. The diffusion coefficient is determined by the Stokes–Einstein equation:(2)$$D=\frac{K_{B}T}{6\pi \eta r}\;$$
where $K_{B},\;T,\;\eta ,\;\mathrm{and}\;r$ represent Boltzmann’s constant, absolute temperature, viscosity, and particle radius, respectively [26]. Every factor was fixed under certain culture conditions. Thus, it is necessary to employ another approach of using convection-diffusion. It is well known that the mentioned gradient disappeared owing to convection in previous studies [27]. However, conventional flow on the surface of cell tissues results in shear stress, which may change the function of cell tissue, and may even cause morphological changes [28,29,30,31]. To realize a culture system in which cultured tissues are continuously provided with fresh media that can be broadly applied to research, simple replacement of the media is the only factor that should be considered. However, if mechanical stimulation that changes the cell function occurs in the process, it will be very difficult to understand the effect on the system. In summary, it is necessary to develop a cell tissue culture system that can replace the media around the tissue without causing mechanical stress-based functional/morphological alterations.

In this study, we developed a culture system in which culture media were perfused into the scaffold of cell tissue with a negligible mechanical stimulus. In this system, a 3D cell tissue made of piled cell sheets was cultured on a porous collagen gel scaffold with a perfusable channel (Figure 1D). Owing to this system, cell tissue viability and glucose metabolism were enhanced without dramatic mechanical stimulus, causing a change in the cell function through mechanotransduction, as previously reported [28,29,30,31]. The present study provides a simple but versatile cell tissue culture system that improves the microenvironment of cultured cell tissues for broad applications in bioengineering.

## 2. Results

### 2.1. System Development

We developed a cell tissue culture system to enhance the abovementioned exchange, as shown in Figure 2. The proposed system is composed of a ring pump (RE-F100, Aquatech Co., Ltd., Osaka, Japan), a bubble trap, a culture vessel with a culture device, and connective tubes. The culture device has a porous collagen scaffold with a perfusable channel (see Appendix A for information of the collagen used), where the cell tissue can be adhered to and cultured (Figure 2B). The ring pump delivers the culture medium. The bubble trap prevents bubbles from entering the channel because without the bubble trap, bubbles entering the channel may prevent the perfusion of the media. The culture media can be delivered into the channel in the collagen gel, which enhances the transport of nutrients and waste. The detailed procedure to prepare the collagen in the device is shown in Appendix A, and the device used in this system was developed in a previous study [32]. Furthermore, in the present study, normal human dermal fibroblasts (NHDF) were employed as model cell types to demonstrate our proposed concept.

### 2.2. Viability of Tissue Cultured Using Each Method

As shown in Figure 3 and Figure A2 in Appendix B, the 3D cell tissues cultured on a general dish underwent damage even with delamination after a three day culture, whereas tissues on collagen scaffolds did not. Since it has been reported that a three day-cultured cell tissue composed of three-layered-cell sheets shows decreased viability, a three day culture was employed in our study to investigate the effect of the proposed culture system [25]. This demonstrates that culture on the porous collagen gel scaffold prevents necrosis inside 3D tissues cultured on a general dish. Figure 4 shows the lactate dehydrogenase (LDH) level in the supernatant cultured with tissue for three days and culture media incubated for three days. LDH has been used as an index to determine cell viability [33]. As shown in Figure 4, except for the tissue cultured on a dish, LDH levels were lower than those in the incubated culture media. In summary, tissues cultured on collagen scaffolds did not secrete LDH compared to general cultured tissues, which is consistent with and supports the qualitative results of hematoxylin and eosin (HE) staining without necrotic cells during culture on the collagen gel (see Figure 3B–D).

### 2.3. Glucose Metabolism

Glucose and lactate concentrations in the supernatant were cultured with tissue for three days, and the culture media were incubated for three days, as shown in Figure 5. The density of glucose in the supernatant decreased with every cell tissue after three day culture, compared with the culture media. Further, there is a certain trend of decreasing glucose along with increased perfusion rate, although this is not clear. In our experiments, the tissue cultured on a dish consumed the minimum amount of glucose, and glucose consumption of tissues cultured on a collagen gel increased along with an increase in perfusion. Figure 5B shows that lactate production from cultured tissues followed a similar trend to that of glucose consumption. In summary, the tissue cultured on a dish produced the least lactate, and lactate production of tissues cultured on a collagen gel increased along with an increase in perfusion late in our experimental conditions.

### 2.4. Gradient of Oxygen Concentration

The gradient of oxygen concentration along the vertical direction (*z*-axis in Figure 6A and Figure A3) above and below the 3D cell tissue cultured on collagen gel was measured under static conditions without perfusion (0 mL/min) and with a perfusion rate of 0.5 mL/min. As shown in Figure 6B, without perfusion, a monotonous decrease was observed in the depth direction of the *z*-axis. With perfusion, different trends in oxygen distribution were observed. As shown in Figure 6C, with perfusion, the minimum oxygen concentration was at the center of the cell tissue. Furthermore, when the results of oxygen distribution with and without perfusion overlapped, the value of oxygen concentration at the center of the cultured tissue was equivalent, as shown in Figure 6D. Furthermore, at the upper side of the cell tissue, the distribution of oxygen concentration showed a similar trend.

### 2.5. Morphological Evaluation of Cell Tissues from the Top View

Top views of 3D tissues before and after three day culture under all conditions in the present study were observed, as shown in Figure 7A. No abnormal morphological changes, such as pierced holes, were observed for all conditions. Further, the shrink ratio of cell tissue cultured for three days under each condition was evaluated, as shown in Figure 7B. The standard deviation of the shrink ratio of the tissue cultured on a dish was larger than that under the other conditions, but except for that there was no clear trend.

### 2.6. Mechanical Stress on Cell Tissue Owing to Perfusion

The deformation of the cell sheet along the *z*-axis due to perfusion with a perfusion rate of 0.5-mL/min was evaluated, as shown in Figure 8. The deformation of cell tissues was measured using a laser displacement meter (ZX1-LD50A61, OMRON, Kyoto, Japan), as shown in Figure A3, after one day static culture. The entire cell tissue was lifted owing to perfusion, and the maximum value of elevation was approximately 0.05 mm.

The shear stress on the surface of the cultured tissue was evaluated using fluorescent particles (R0300, Thermo Scientific, Waltham, MA, USA). The density and volume of the particle were 1.05 g/cm^3^ and 14.1 µm^3^, respectively. Based on the experimental results, no moving particles were observed. This observation was performed after one day culture under static conditions, and the employed perfusion rate was 0.5 mL/min.

## 3. Discussion

We hypothesized that culturing cell tissue on porous collagen gel would improve the cell culture environment owing to the media provided from the bottom of the cell tissue and that perfusion causes convection-diffusion and an improved culture environment compared with static culture. This hypothesis is based on the convection-diffusion of culture media that should be realized in our proposed system. The above hypothesis was supported by the results (see Figure 6), and we proposed that fresh media could be provided to cultured 3D tissue in our system. As shown in Figure 3, the 3D tissue cultured on a general dish exclusively showed necrosis inside the tissue, which occurs because of diffusion limitations [34]; thus, we concluded that culturing cell tissue on the porous collagen gel scaffold improved the diffusion rate even inside the 3D tissue. The advantage of perfusion can be determined from glucose and lactate assays as well as the evaluation of the oxygen gradient. The results of the glucose and lactate assays showed a trend of increased glucose metabolism along with an increase in the perfusion rate. This indicates that glucose-rich media were provided to the tissue owing to perfusion or that the transportation of glucose improved cell tissue viability. Although only the glucose intake and lactate output were evaluated in this study, the intake and output of other substances are expected to be enhanced by the present system. Furthermore, as shown in Figure 8, there was a convection flow in the collagen scaffold owing to perfusion. The evaluation of the oxygen gradient also indicates the effect of perfusion. Therefore, owing to the perfusion in the collagen scaffold, there was a convection flow in the scaffold that provided fresh media to the cultured tissue.

The oxygen gradient was particularly interesting from three viewpoints. First, as shown in Figure 6B, without perfusion, the oxygen concentration was lower below the cell tissue than that inside the tissue. This can be explained by the one-dimensional (1D) convection-diffusion Equation:(3)$$\frac{\partial \phi }{\partial t}+v\frac{\partial \phi }{\partial z}-D\frac{\partial^{2}\phi }{\partial z^{2}}=S\left(z\right)$$
where v, z, and S(z), respectively represent the velocity of the convection flow, the location along the *z*-axis in Figure 6A, and the source term [35]. The gradient was measured at one time point, and there should be no gradient if *t* = 0. Thus, ϕ is a function of *z*. Under static conditions, *v* = 0. Furthermore, the cell sheet *S* has a negative value when *z* = 0. However, at a certain position above the cell sheet, *S* has a positive value because of the interface between the media and air, while *S* is 0 below the cell sheet owing to the bottom of the vessel. Under the initial condition before culturing the cell tissue, the oxygen concentration should be homogeneous [22], and thus this monotonous decrease in the oxygen gradient is reasonable. Although the oxygen gradient around the tissue cultured on the dish was not measured, *S* should have negative and positive values at the cell tissue and the interface between the media and air, respectively. More oxygen exists in the atmosphere than in the liquid, even if the liquid is oxygen-saturated [36]. Thus, the interface between the medium and air is a source term. Note that oxygen is provided only from the air-liquid interface in our system. It is reasonable to regard the surface source as a volume source at an infinitesimal volume, as has been numerically done in previous research [37]. On the other hand, under conventional culture conditions using a general culture dish, *S* = 0 below the cell tissue even at *t* = 0, which indicates a more hypoxic condition. Second, with perfusion, the minimum value was observed inside the tissue, and oxygen concentration below the cell tissue was increased due to the perfusion compared to a condition without perfusion. Therefore, it can be predicted that the source term is also located below the cell tissue. This source term should be the perfused microchannel, that is, *S* has a positive value at certain points above and below the cell tissue. Furthermore, as shown in Figure 8, leaky flow from the microchannel pushes cell tissues upward, which indicates that *v* has a positive value with perfusion. In short, the import of oxygen was definitely increased owing to perfusion. Farther, even without perfusion, the porous collagen scaffold imports more oxygen than the conventional dish culture, since from the lateral side of the collagen gel fresh media was provided to cell tissues. Third, even with perfusion, the oxygen concentration inside the cell tissue was similar to that without perfusion. Moreover, as the source values increase, it can be assumed that the loss term increases. In other words, the oxygen imported to and consumed by the cell tissue increases with perfusion. Inside the dense tissues, convection flow cannot be expected, and the diffusion term dominates the transport. However, as shown in Equation (2), it is difficult to change the diffusion coefficient *D* for the cell culture conditions. Thus, to transport oxygen into the tissue, the oxygen concentration at the surface of the tissue should be increased, which has been realized in our proposed system.

We also evaluated the mechanical stimulus applied to the cell tissues owing to the present culture method, because cell function can be regulated by mechanical stimulation [38,39,40,41]. It is known that cultured tissue based on cell sheet technology shrinks during culture owing to its active inter/intracellular tension [42] because the tissue does not have any artificial inclusion to disturb the deformation. Moreover, dramatic deformation of cell tissue affects cell function [43]; hence, the shrinkage rate of the cell tissue cultured under each condition was evaluated. Regardless of the perfusion rate, no trend was observed in the shrinkage rate. One of the driving forces for intracellular tension was actin skeleton, which can be affected by mechanical stimulation [44,45,46]. Thus, this result indicates mechanical stimulation affecting actin polymerization is not given in our culture system. Furthermore, the shrinkage rates of the tissues cultured on the collagen scaffolds were more stable than those cultured on general dishes, which may indicate the good stability of the present culture method owing to the stable adhesion between the cell tissue and collagen gel. We also showed that the lifting of the cell tissue was entirely owing to perfusion. As the level of lifting at the measured points of the sheet was not homogeneous, the cell tissue was stretched. In our proposed system, even if we focused on the most stretched part, there was a deformation of 0.017%. However, in previous studies, approximately 68% stretch was employed to change the cell sheet morphology dramatically [13]. Furthermore, owing to the intra/intercellular force, more than 10% of deformations occurred during the three-day stationary and perfusion cultures. Thus, the deformation caused by perfusion should be minor. Observing particles did not confirm the convection flow, as shown in Figure A3, which is consistent with a comparable oxygen gradient above the cell tissue, with and without perfusion. Consequently, we argue that there is a negligible microenvironment from the perspective of the mechanical stimulus in our system.

We developed a system in which 3D cell tissues can be provided with fresh culture medium. The greatest contribution of our novel system is the realization of a 3D cell culture method to maintain cell viability higher than that of a conventional cell culture method using a general dish. Furthermore, although we evaluated glucose intake, lactate production, and oxygen gradient in this study as representative factors, the intake, output, and gradient of other substances might be improved. However, mechanical stimulation resulting in major functional and morphological changes in the cultured tissue has not been confirmed. Although the cell tissue of three-layered-NHDF cell sheets was used to investigate the proposed concept, our system does not have a strict limitation on the cell type or the number of layered cell sheets. It is expected that perfusion may affect the viability of tissues if other cell types with higher metabolism or cell tissues with more layered cell sheets are employed in this system. The fabrication of 3D cell tissue is a very common goal in tissue engineering, which is a basic technology for regenerative medicine, cell-based food, and drug screening assay models [1,2,3]. Furthermore, although the scaffold was made of native collagen gel, another type of biocompatible porous gel could be utilized. Thus, when implanting the fabricated 3D cell tissues to a patient, an atelocollagen gel can be used. For the fabrication of cell-based food, this collagen gel is of biological origin and has no toxicity, which indicates the possibility of eating the tissue with a scaffold. Since other types of gel can be used as a scaffold, investigations into the type of gel may be an interesting topic to realize a better culture environment. Although we used our system to culture a 3D cell tissue made of layered cell sheets, it can be employed to culture any form of cell tissue. As the culture environment around cell tissues can be improved, a monolayer cell sheet, cell aggregation, or organoids may be a target. Therefore, our proposed system can be a fundamental technology in the field of tissue engineering and should advance the broad field of bioengineering.

## 4. Materials and Methods

### 4.1. Cell Culture

NHDF (CC-2511, Lonza, Basel, Switzerland) cells were cultured in Dulbecco’s Modified Eagle Medium (D-MEM; 043-30085, FUJIFILM Wako Pure Chemical Corporation, Osaka, Japan) supplemented with 10% fetal bovine serum (FBS) (10270, Thermo Fisher, Waltham, MA, USA) and 1% penicillin-streptomycin (168-23191, FUJIFILM Wako Pure Chemical Corporation, Osaka, Japan). Cells were harvested using 5% trypsin (208-17251, FUJIFILM Wako Pure Chemical Corporation, Osaka, Japan). Then, 2 × 10^6^ cells were seeded on a temperature-responsive cell culture dish (UpCell^TM^ dish, I.D. 35 mm, Type-E, CellSeed, Tokyo, Japan), the surface of which was coated with a matrix coating (Easy iMatrix-511, 892018, Takara Bio Inc., Shiga, Japan). To obtain the coated surface, Easy iMatrix-511 was introduced into the dish, which was then incubated for 1 h at 37 °C. The seeded cells were cultured for three days. Cells were harvested as a cell sheet by incubating the culture dish in a CO_2_ incubator set at 20 °C for 30 min.

### 4.2. Morphological Evaluation

To obtain a cross-sectional view of the cell tissue, the cell tissue was cut around the center. Hematoxylin and eosin staining was then performed using the conventional method. To evaluate the shrinkage rate, as shown in Figure 7B, the areas of cell sheets before and after three days of culture were measured manually using Image J (Bethesda, MD, Rockville, USA) [47].

### 4.3. Oxygen Gradient Evaluation

To evaluate the oxygen gradient, the concentration of oxygen in the culture media was measured using an oxygen sensor (OX-100, Aarhus, Denmark). For the measurement, after 1-day static culture, the entire system was set in the chamber (Invivo_2_ 300, Ruskinn, Bridgend, UK). The media was agitated, and the oxygen gradient was affected. Thus, 1 h before the measurement, gentle pipetting was applied to match the conditions. 

## Figures and Tables

**Figure 1 ijms-22-06780-f001:**
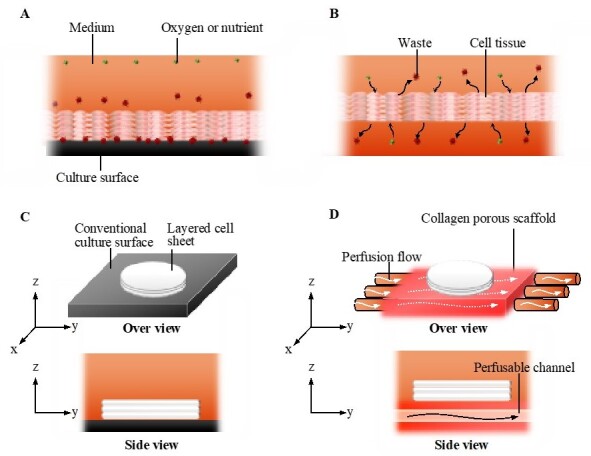
The concept of the perfusion system. Comparison of conventional culture (**A**) and the ideal culture (**B**) environment. Schematic images of the conventional culture method (**C**) and the culture method realized in this study (**D**).

**Figure 2 ijms-22-06780-f002:**
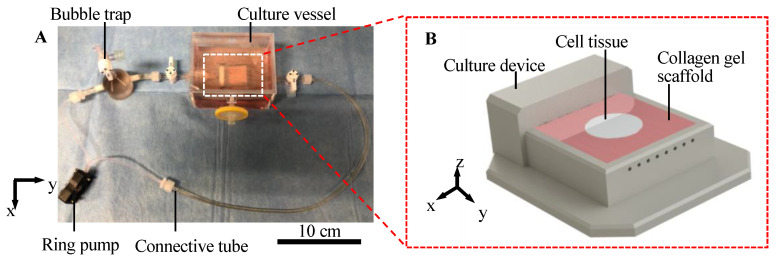
Illustration of the culture system. Overview of the whole perfusion system (**A**). Schematic image of culture device with the collagen scaffold (**B**).

**Figure 3 ijms-22-06780-f003:**
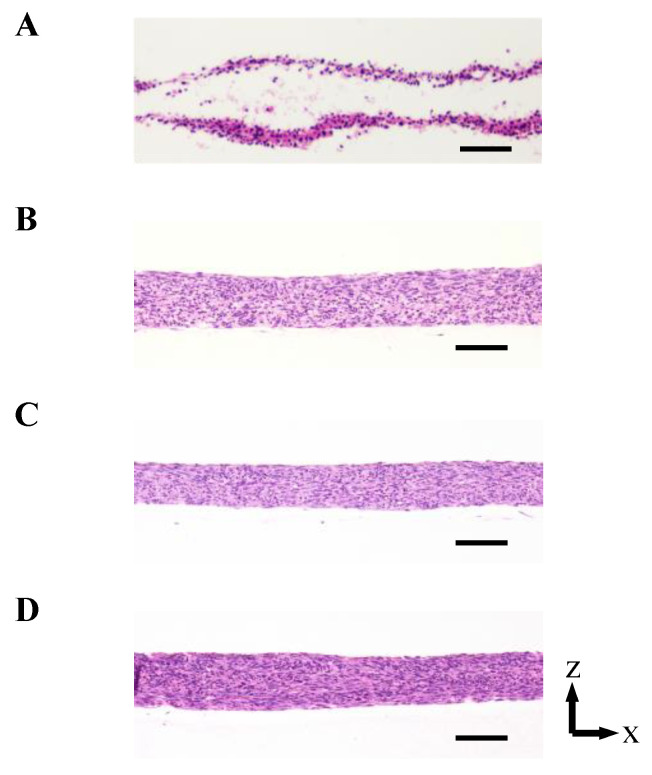
The densities of the cell tissue cultured for 3 d with each method. HE stained cross-section views with the cell tissue culture on a general dish (**A**) and collagen gel with perfusion rates of 0 (**B**), 0.3 (**C**), and 0.5 (**D**) mL/min. Scale bars indicate 100 µm. Figure A2 in Appendix B shows the reproducibility of the results with the cell tissue culture on a general dish and collagen gel with a 0.5-mL/min perfusion rate.

**Figure 4 ijms-22-06780-f004:**
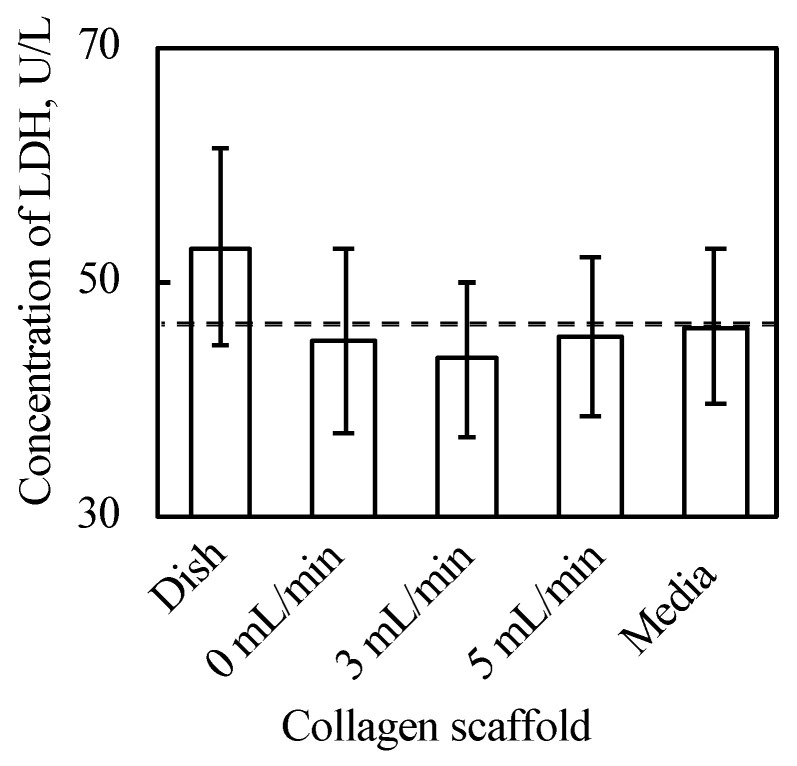
Results of LDH assay (*n* = 3~7, mean ± SD). The cell tissue cultured on ubiquitous culture dish released LDH in the media. However, cell tissues cultured using the other methods did not show the release.

**Figure 5 ijms-22-06780-f005:**
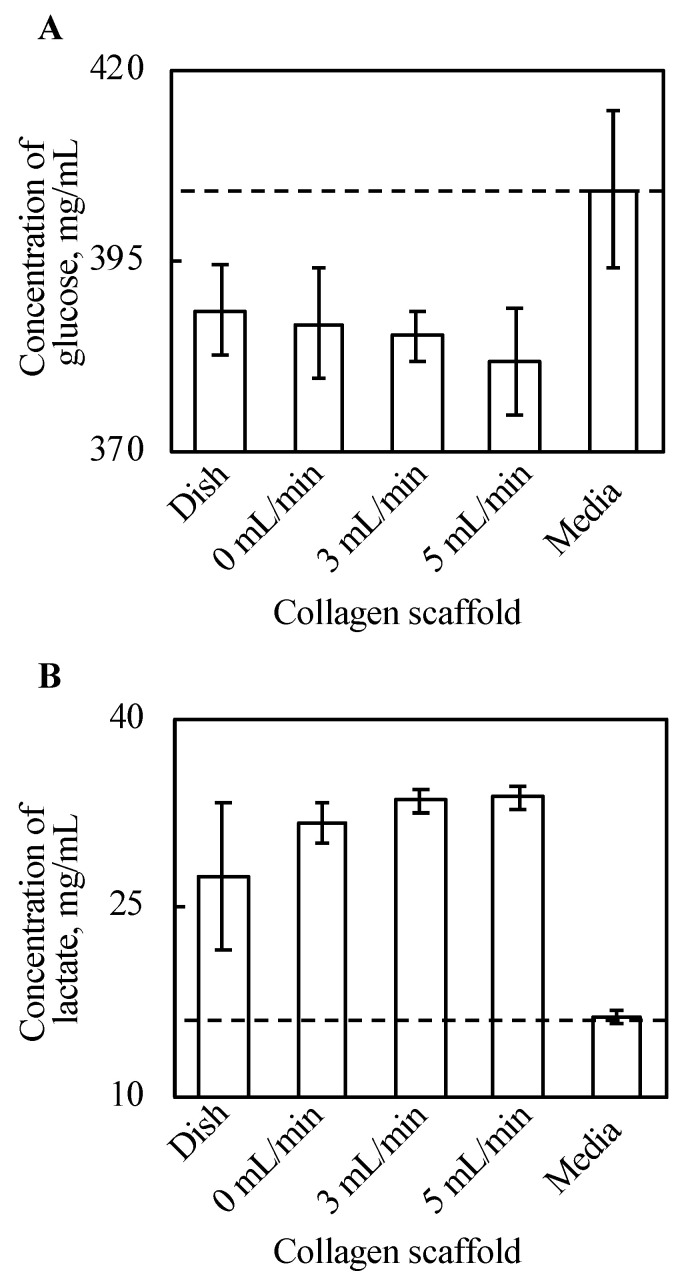
The results of glucose assay (**A**) and lactate assay (**B**) (*n* = 3~7, mean ± SD). The *y*-axis represents the concentration of each substance in the supernatant with each culture method and incubated DMEM without cells. The cell tissue cultured on a collagen gel scaffold shows a higher glucose consumption and lactate production. Further, along with increasing perfusion speed, both the glucose consumption and lactate production increased.

**Figure 6 ijms-22-06780-f006:**
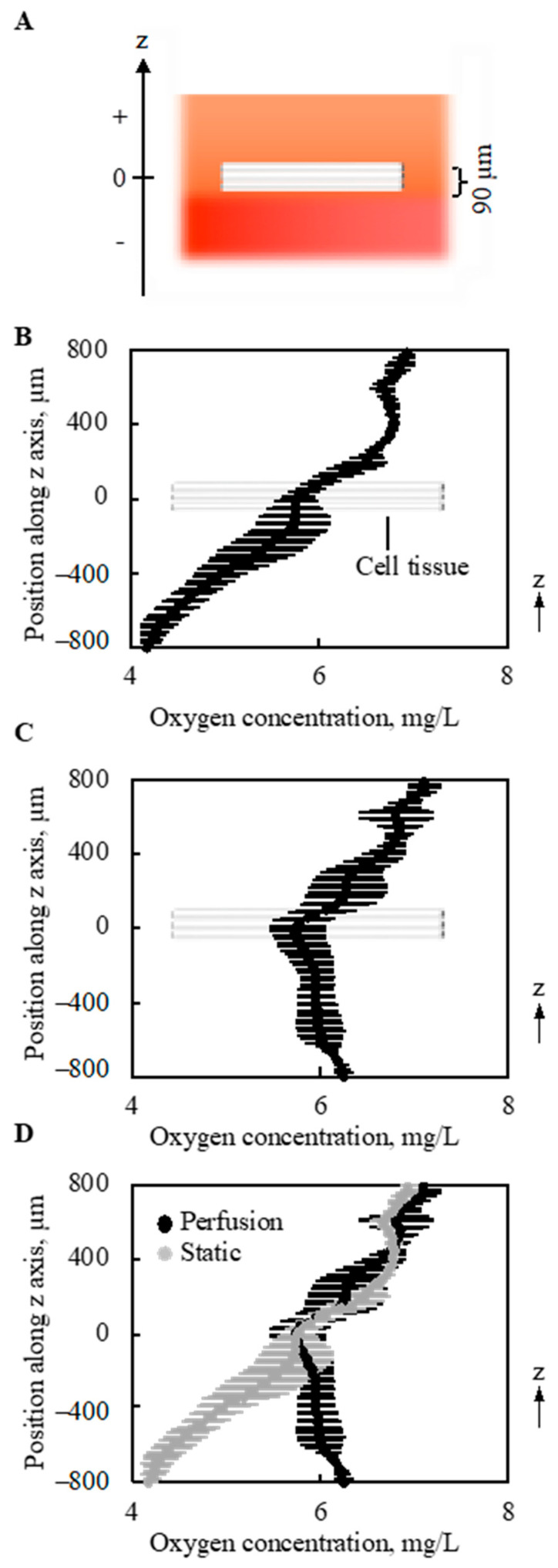
Oxygen concentration around and in the tissue (*n* = 3). The schematic image showing measured region (**A**). Without perfusion, a gradual decrease in the oxygen concentration was shown along with the depth direction (**B**). However, owing to the perfusion, the minimal point of oxygen concentration was at the location of the cell tissue (**C**). On the other hand, the density of oxygen itself around the cell tissue is very similar. The results shown in (**B**,**C**) were overlapped for comparison (**D**). Note that the perfusion speed is 5 mL/min.

**Figure 7 ijms-22-06780-f007:**
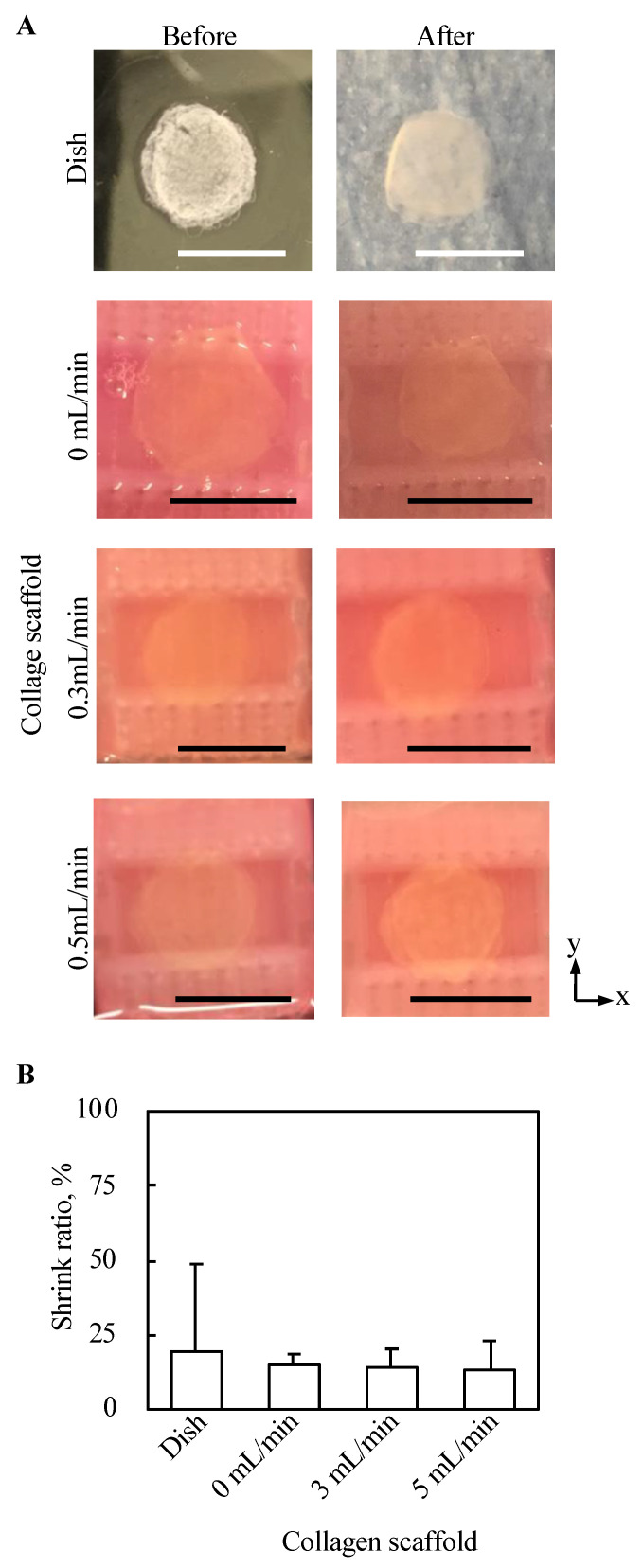
Evaluation of the morphology from the top view of tissue cultured using each method. The macro top-view images of the tissues (**A**). The shrink ratios of tissues are shown (*n* = 3~7, mean ± SD) (**B**).

**Figure 8 ijms-22-06780-f008:**
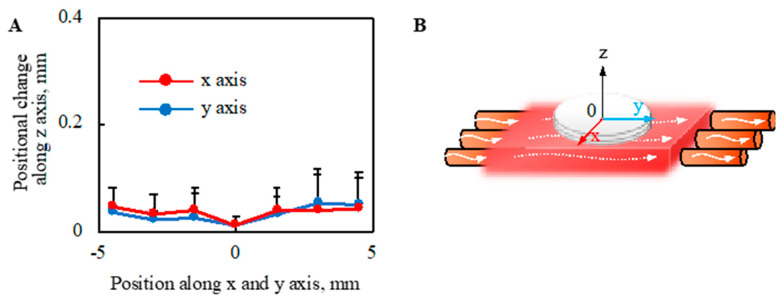
Evaluation of morphological change in the direction of the z axis owing to the perfusion. The positional change of the cell tissue surface owing to perfusion (**A**) is shown (*n* = 3) (details are shown in Figure A3). Note that the schematic image to show each axis is shown (**B**). Note that the perfusion speed was 5 mL/min in this figure.

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
