# Peer review of "Perfusable System Using Porous Collagen Gel Scaffold Actively Provides Fresh Culture Media to a Cultured 3D Tissue"

_ijms, 2021, doi:10.3390/ijms22136780_

Round 1

Reviewer 1 Report

I have read the manuscript and I find it quite appealing and very interesting. I believe that the authors should add some references in some points, correct some parts and respond to some questions that I have. Other than that the manuscript can be accepted. I would like to see the responses of the authors before full acceptance

See pdf attached with the comments

Reviewer 2 Report

The authors introduced a novel system that employs diffusion and convection to enhance transportation. The introduced culturing system can be a promising alternative standard for 3D cell culture. The study is innovative and the results are well-interpreted.  

Regarding the collagen scaffold, the authors should define the type of collagen and the physical characterization of the obtained scaffold (porosity, free volumes). The macro and microporosity (free volume and free volume distribution)  could be of impact in the course of the standardization and optimization of the 3D cell culture model.

Round 2

Reviewer 2 Report

The authors adequately answered the reviewer's comments and modified the paper accordingly. I suggest the publication without further revision.